# Phlorizin Protects Against Oxidative Stress and Inflammation in Age-Related Macular Degeneration Model

**DOI:** 10.3390/biom15040523

**Published:** 2025-04-03

**Authors:** Zhen-Yu Liao, Chih-Yu Hung, Yu-Jou Hsu, I-Chia Liang, Yi-Chun Chen, Chao-Hsien Sung, Chi-Feng Hung

**Affiliations:** 1Department of Internal Medicine, Shin Kong Wu Ho-Su Memorial Hospital, Taipei 111, Taiwan; m011950@ms.skh.org.tw; 2Department of Ophthalmology, Chang Gung Memorial Hospital, Kweishan, Taoyuan 333, Taiwan; 3PhD Program in Pharmaceutical Biotechnology, Fu Jen Catholic University, New Taipei City 242, Taiwan; 411138028@m365.fju.edu.tw (Y.-J.H.); 157500@mail.fju.edu.tw (Y.-C.C.); 4National Defense Medical Center, Department of Ophthalmology, Tri-Service General Hospital, Taipei 114, Taiwan; doc30826@mail.ndmctsgh.edu.tw; 5Division of Anesthesiology, Fu Jen Catholic University Hospital, Fu Jen Catholic University, New Taipei City 242, Taiwan; 6School of Medicine, Fu Jen Catholic University, New Taipei City 242, Taiwan; 7School of Pharmacy, Kaohsiung Medical University, Kaohsiung 807, Taiwan

**Keywords:** phlorizin, age-related macular degeneration, retinal pigment epithelial cells, inflammation, oxidative stress

## Abstract

Background: Sweet Tea (Lithocarpus polystachyus Rehd.), a traditional ethnobotanical medicine, contains phlorizin, a dihydrochalcone compound with antioxidative and anti-inflammatory properties. Given the critical role of oxidative stress and inflammation in age-related macular degeneration (AMD), this study tested the hypothesis that phlorizin mitigates oxidative damage and inflammation in AMD models, thereby offering therapeutic potential. Materials and Methods: Adult retinal pigmented epithelial cells (ARPE-19) were pre-treated with phlorizin (0.01–0.1 μM) and subjected to oxidative stress induced by ultraviolet A (UVA) radiation or sodium iodate (NaIO_3_). Cell viability, reactive oxygen species (ROS) production, MAPK/NF-κB signaling, and the level of pro-inflammatory cytokines (IL-1β, IL-6, TNF-α) and pro-angiogenic factors (VEGF, MMP2, MMP9) expression were assessed using MTT assays, fluorescence imaging, Western blotting, and RT-qPCR. In vivo, a laser-induced choroidal neovascularization (CNV) mouse model was used to evaluate phlorizin’s effects on CNV formation and vascular leakage via fundus photography and fluorescence angiography. Result: Phlorizin significantly enhanced cell viability, reduced ROS production, inhibited MAPK/NF-κB activation, and downregulated inflammatory and angiogenic mediators. In vivo studies confirmed the reduced CNV formation and vascular leakage following the phlorizin treatment. Conclusions: Phlorizin demonstrated significant protective effects against oxidative stress and inflammation, highlighting its therapeutic potential for treating AMD.

## 1. Introduction

Age-related macular degeneration (AMD) is a long-term, progressive neurodegenerative condition and a major cause of permanent vision loss and blindness globally [1,2,3]. According to the World Health Organization (WHO), AMD is the third most common cause of blindness globally, affecting millions of individuals and imposing a significant burden on health systems [4,5]. AMD is recognized as a multifactorial disease, with its onset influenced by several factors, including aging, diet, smoking, hereditary factors, race, cardiovascular diseases, alcohol consumption, obesity, and overexposure to bright light [3,6,7,8,9,10,11,12,13,14]. The prevalence of AMD is expected to increase substantially due to the aging population and is estimated to affect 288 million people by 2040 [2,10,15,16]. The spectrum of AMD spans from neovascular AMD (wet AMD) to geographic atrophic AMD (GA AMD or dry AMD) [17]. In the early stages of dry AMD, patients often exhibit drusen deposits located between the retinal pigment epithelium (RPE) basal lamina and Bruch’s membrane [18]. Geographic atrophy (GA) AMD is identified by pale, atrophic lesions in the outer area of the retina, reflecting the gradual loss of the RPE, choriocapillaris, and photoreceptors [19]. Approximately 15% of AMD patients present with neovascular AMD. The pathognomonic feature of wet AMD is the choroidal neovascularization (CNV) beneath the neurosensory retina, which often leads to fluid or blood leaking into the subretinal space [20]. The overproduction of vascular endothelial growth factors (VEGFs) is the key feature of CNV [21]. Although treatments for wet AMD, such as photocoagulation therapy and anti-VEGF injections, have advanced, they rarely achieve complete remission, and recurrence is common [22,23]. Furthermore, these treatments neither halt the disease progression in its early stages nor treat dry AMD, for which no approved therapies exist. There remains an urgent need for novel preventive and therapeutic strategies for both forms of AMD [24,25].

Aging and environmental factors, particularly ultraviolet (UV) radiation, play significant roles in the AMD development [26,27,28]. UV exposure leads to an overproduction of reactive oxygen species (ROS), overwhelming the antioxidative defense mechanisms in retinal pigment epithelium (RPE) cells. This causes oxidative stress, which damages photoreceptors, RPE cells, and the choriocapillaris [12,26,29,30,31]. When the skin is exposed to UV radiation, it activates key signaling pathways, including the nuclear factor kappa-light-chain-enhancer of activated B cells (NF-κB) and mitogen-activated protein kinases (MAPK). This activation leads to an increased expression of pro-inflammatory cytokines, such as interleukin-1β (IL-1β), IL-6, and tumor necrosis factor- α (TNF-α), as well as angiogenic factors, like vascular endothelial growth factor (VEGF) [32,33,34]. Previous studies have shown that UV exposure can activate the MAPK pathways—specifically ERK, JNK, and p38—in human RPE cells [35,36]. Additionally, the NF-κB pathway could be activated by photooxidative stress [37,38].The oxidative damages inflicted on RPE cells, along with the subsequent disruption of the blood–retina barrier (BRB), may further contribute to the AMD pathogenesis [39].

Sodium iodate (NaIO_3_), a stable oxidizing compound, can selectively damage RPE cells and is utilized in oxidative stress models for studying retinal neurodegeneration [40,41]. In the melanocytes of the RPE cells, NaIO_3_ transforms glycine into potentially harmful glyoxylate and suppresses enzyme activities [42]. The NaIO_3_- induced retinochoroidal damage is very similar to that observed in dry AMD [6,7,43,44]. NaIO_3_ generates ROS, damages RPE cells, disrupts the BRB, and affects the choriocapillaris [45,46,47,48,49]. Studies show that NaIO_3_ induces apoptosis, necroptosis, and pyroptosis in ARPE-19 cells, as evidenced by Caspase activation, receptor-interacting protein kinase-3 (RIPK-3) aggregation, and high mobility group box-1 (HMGB-1) release [7,50,51]. While NaIO_3_-induced RPE death does not fully explain the mechanisms of the GA form of AMD, it offers insights into RPE damage and its impact on the choriocapillaris. Collectively, these mechanisms lead to RPE cell death, which subsequently causes photoreceptor degeneration and vision impairment, making NaIO_3_ a widely used model for studying retinal diseases in preclinical research [52].

Phlorizin, a flavonoid compound derived from sweet tea [53] or the bark of apple trees [54], has garnered attention for its diverse pharmacological properties, including antioxidative, anti-inflammatory, anti-tumor, and neuroprotective effects. Liu et al. demonstrated that phlorizin mitigates oxidative stress in aging models induced by D-galactose, highlighting its role in combating oxidative damage and exerting neuroprotective effects [55]. Ma et al. showed that phlorizin reduces the oxidative injury caused by exhaustive exercise in mice through the Nrf2/ARE signaling pathway [56]. Research by Cambeiro-Pérez et al. (2021) explored its immunomodulatory activities on human THP-1 macrophages, revealing that phlorizin alters metabolic pathways to reduce inflammation [57]. Phlorizin also inhibits the progression of cancer cells’ proliferation by the JAK2/STAT3 pathway [58]. Previous studies have highlighted its potential in preventing diabetic retinopathy and other oxidative stress-related conditions [59]. The diet supplement of such compounds could help eliminate free radicals, boost the function of antioxidant enzymes like superoxide dismutase (SOD) and catalase (CAT), and inhibit lipid peroxidation [60]. These unique properties of flavonoid compounds have positioned them as potential alternative therapeutic strategies for treating various neurodegenerative diseases, particularly Alzheimer’s disease [61,62]. However, its therapeutic role in AMD remains unexplored. Given its potent antioxidative capabilities and ability to modulate inflammatory responses, phlorizin presents a promising candidate for treating AMD or preventing disease progression. We aim to investigate the beneficial effects and possible mechanisms of phlorizin in mitigating damages caused by UVA and NaIO_3_ in adult retinal pigment epithelial cells (ARPE19) in this research. The findings may pave the way for developing novel strategies to combat this debilitating ocular disease, ultimately improving the quality of life for affected individuals.

## 2. Materials and Methods

### 2.1. Phlorizin

Phlorizin was obtained from Cayman Chemical, located in Ann Arbor, MI, USA.

### 2.2. Cell Line Culture

In this study, the ARPE-19 cell line was utilized. These cells were obtained from the Biological Resource Collection and Research Center (BCRC) at the Food Industry Research and Development Institute, under the catalog number BCRC60383. Being adherent in nature, the cells were cultured in a humidified incubator maintained at 37 °C with 5% CO_2_. The culture medium used was Dulbecco’s Modified Eagle Medium (DMEM), containing 1% antibiotic–antimycotic (AA) solution and 10% fetal bovine serum (FBS). Subculturing was performed once the cells reached a confluency level of 80–90%.

### 2.3. UVA Source

UVA exposure was administered utilizing Bio-Sun system illuminator (Vilber Lourmat, Collégien, France). The configuration consisted of four lamps positioned to ensure uniform illumination at a distance of 10 cm. Ultraviolet radiation was emitted within the 355–375 nm spectrum, with peak intensity at 365 nm. To achieve a UVA dose of 20 J/cm^2^, with an irradiance of 4.2–4.5 mW/cm^2^, an exposure duration of approximately 70–80 min was necessary. The Bio-Sun system, equipped with a programmable microprocessor, automatically ceased UV emission upon reaching the specified energy level.

### 2.4. MTT Assay

Cells were plated in a 24-well plate at a concentration of 10 × 10^4^ cells per well. After being exposed to phlorizin, UVA or both, 300 μL of MTT solution [3-(4,5-dimethylthiazol-2-yl)-2,5-diphenyl-2H-tetrazolium bromide] was introduced into each well. The plate was then incubated at 37 °C for approximately 2 h. Following incubation, the medium was discarded, and the resulting formazan crystals were solubilized using dimethyl sulfoxide (DMSO). After thorough mixing, cell viability was assessed by measuring absorbance at 550 nm using an enzyme-linked immunosorbent assay (ELISA) reader.

### 2.5. Measurement of ROS

ROS were measured utilizing the fluorescent probe DCFH-DA (2′,7′-dichlorodihydrofluorescein diacetate). Once inside the cell, this compound is hydrolyzed by intracellular esterases into the non-fluorescent form, DCFH. Subsequently, DCFH is oxidized by ROS to generate fluorescent DCF. The fluorescence intensity was then determined using either flow cytometry or fluorescence microscopy.

### 2.6. Western Blot Analysis

To investigate changes in intracellular protein levels, Western blotting was conducted. ARPE-19 cells were cultured in 10 cm dishes until they reached 90% confluence, followed by a 24 h period of nutrient deprivation. The cells were then pre-treated with phlorizin for 24 h before being exposed to UVA radiation at a dose of 20 J/cm^2^ or NaIO_3_. After treatment, the cells were collected by scraping, lysed through sonication, and centrifuged at 13,200 rpm for 10 min at 4 °C. Concentration of protein retrieved from supernatant was determined using the Pierce protein assay kit (Pierce, Rockford, IL, USA). Proteins were separated on a 10% SDS-polyacrylamide gel via electrophoresis and transferred onto a PVDF membrane using an electroblotting method. The PVDF membrane was placed in a TBS–T solution (Tris-buffered saline with 0.05% Tween 20) containing 5% skimmed milk powder and shaken for 1 h to block non-specific binding. The membrane was rinsed three times with TBS–T, each wash lasting 10 min. It was incubated overnight at 4 °C after adding primary antibodies, diluted at a ratio of 1:1000. The membrane was washed three more times with TBS-T for 10 min per wash. Secondary antibodies, also diluted to 1:1000, were added and left for 1 h, after which the membrane underwent another three rounds of washing with TBS-T, each lasting 10 min. Finally, a developer solution was applied, and the membrane was analyzed using a chemical luminescence imaging system (BIOSTEP Celvin^® ^, biostep, Burkhardtsdorf, Germany). Western blot original images can be found in Appendix A.

### 2.7. Real-Time Quantitative Polymerase Chain Reaction (RT-qPCR) Analysis

ARPE-19 cells were plated in culture dishes and cultured until they reached 90% confluence, followed by a 24 h starvation period. Subsequently, the cells were pre-treated with phlorizin for 24 h and then exposed to UVA radiation at a dose of 20 J/cm^2^ or NaIO_3_. The cells were collected by scraping and centrifuged at 16,000× *g* for 10 min at 4 °C. RNA was isolated using a total RNA extraction kit (GeneDireX^®^, Vegas, NV, USA) from the supernatant. Based on the protocol of the iScript™ cDNA Synthesis Kit (BIO-RAD, Hercules, CA, USA), cDNA was synthesized from RNA. PowerUp™ SYBR™ Green Master Mix (Applied Biosystems™, Waltham, MA, USA) was used. The mixture included 7.5 μL of ddH2O, 2 μL of cDNA, 0.25 μL each of forward and reverse primers, and 10 μL of SYBR GREEN, which were combined thoroughly. Primer sequences are detailed in Table 1. The RNA was then quantified using the ABI StepOnePlus™ Real-time PCR System (Thermo Fisher Scientific, Waltham, MA, USA).

### 2.8. Animals

C57BL/6J mice, aged four to six weeks, were sourced from the National Biotechnology Research Park in Taipei, Taiwan. All animal housing and experimental procedures adhered to the guidelines established by the Fu Jen University Laboratory Animal Center. The mice were divided into three distinct groups: (1) a group subjected to laser treatment alone without phlorizin administration; (2) a group receiving a daily dose of 3 mg/kg phlorizin alongside laser treatment; and (3) a group treated with a laser and supplemented with a daily dose of 10 mg/kg phlorizin.

Induction of general anesthesia was through an intraperitoneal (IP) injection of a mixture containing Zoletil™ 50 (containing zolazepam and tiletamine) and Rompun 20 (xylazine) at a dose of 1 μL per gram of body weight in a 3:2 ratio. Pupil dilation was facilitated using a mixture of phenylephrine hydrochloride at a concentration of 0.5% and tropicamide at a concentration of 0.5% (Santen Pharmaceutical Co., Ltd., Tokyo, Japan) during procedures such as laser treatment, color fundus photography (CFP), and fluorescein angiography (FA).

### 2.9. Laser-Induced CNV Model

On the fifth day, CNV was initiated using laser photocoagulation, as outlined in prior methodologies [63,64]. In summary, mice were anesthetized, and their pupils were dilated prior to the procedure. A 532 nm green laser photocoagulator (LIGHTLas 532, LIGHTMED, San Clemente, CA, USA) was used to administer laser spots (100 μm in diameter, 0.15 s in duration, and 150 mW power) to the retina. The procedure was conducted using a slit-lamp system, with a handheld coverslip serving as contact lens. Four laser burns were applied at the positions corresponding to 3, 6, 9, and 12 o’clock around the head of optic nerve. The successful induction of CNV at each site was confirmed by the appearance of a bubble during the procedure, signifying the rupture of Bruch’s membrane.

### 2.10. Color Fundus Photography (CFP) and Fluorescence Angiography (FA)

On the tenth day, morphological alterations were analyzed by a Micron IV microscope (Phoenix Research Laboratories, Pleasanton, CA, USA). Following anesthesia and pupil dilation, each mouse was placed laterally on the microscope stage, and a 2% Methocel gel (OmniVision, SA, Neuhausen, Switzerland) was used to keep the eye moist. CFP was conducted initially, followed by fluorescein FA. For the FA procedure, 0.05 mL of 10% fluorescein was administered intraperitoneally, and sequential images and videos were captured.

The angiograms were independently evaluated in a blinded fashion by two retinal specialists, Yi-Chia Liang and Yun-Hsiang Chang, using a standardized FA grading system [65]. According to this system, Grade 1 represented the absence of hyperfluorescence; Grade 2 indicated hyperfluorescence without leakage; Grade 3 demonstrated hyperfluorescence in the early or mid-phase with leakage in the late phase; and Grade 4 showed intense hyperfluorescence during the transit phase with leakage extending beyond the treated regions in the late phase. Subretinal hemorrhages during the procedure were rare, and the resulting obstruction of fluorescein visualization during FA occurred infrequently. To minimize potential bias, laser spots impacted by this issue were excluded from FA grading.

### 2.11. Statistical Analysis

Results are expressed as the mean ± standard error of the mean (SEM). Statistical analyses comparing control and experimental groups were conducted using unpaired two-tailed Student’s *t*-tests with SigmaPlot software (Version 14.0). A *p*-value of less than 0.05 was regarded as statistically significant.

## 3. Results

### 3.1. Cell Viability Test

#### 3.1.1. Cytotoxicity of Phlorizin on ARPE-19 Cells

The cytotoxicity of phlorizin on ARPE-19 cells was evaluated using the MTT assay. Cells were pre-treated with various concentrations of phlorizin (0.01 μM, 0.03 μM, 0.1 μM, 0.3 μM, and 1 μM) for 24 h. The results demonstrated that phlorizin at concentrations ranging from 0.01 μM to 1 μM did not induce significant cytotoxicity in ARPE-19 cells, maintaining cell viability above 90% across all tested concentrations (Figure 1A). Based on these findings, concentrations of 0.01 μM, 0.03 μM, and 0.1 μM were selected for subsequent experiments.

#### 3.1.2. UVA Exposure Effect on ARPE-19 Cell Viability

The viability of ARPE-19 cells following UVA radiation at different intensities (5 J/cm^2^, 10 J/cm^2^, 15 J/cm^2^, 20 J/cm^2^, 25 J/cm^2^, and 30 J/cm^2^) was examined. Cell survival was found to decline in a dose-dependent fashion as the UVA exposure increased. Notably, UVA doses of 10 J/cm^2^, 15 J/cm^2^, 20 J/cm^2^, 25 J/cm^2^, and 30 J/cm^2^ impaired cell viability by approximately 18%, 24%, 38%, 47%, and 54%, respectively (Figure 1B). Based on these results and past findings, a UVA exposure at 20 J/cm^2^ was selected for subsequent experiments.

#### 3.1.3. Phlorizin’s Protective Effect Against UVA-Induced Cytotoxicity

The potential protective effect of phlorizin against UVA-induced cytotoxicity was investigated by pretreating ARPE-19 cells with 0.01 μM, 0.03 μM, and 0.1 μM of phlorizin for 24 h, followed by 20 J/cm^2^ of UVA. Results from the MTT assay showed that 20 J/cm^2^ of UVA alone caused about 38% cell death. However, the phlorizin pretreatment significantly enhanced cell viability in a concentration-dependent manner. Notably, at 0.1 μM, the viability recovery reached approximately 85%, highlighting the robust protective effect of phlorizin under UVA radiation (Figure 1C).

### 3.2. Phlorizin Inhibits UVA-Induced ROS Generation

The role of phlorizin in attenuating ROS production from UVA exposure was analyzed using the DCFH-DA fluorescence assay. The pretreatment with phlorizin at 0.01 μM, 0.03 μM, and 0.1 μM significantly inhibited ROS generation in a dose-dependent fashion after UVA radiation at an intensity of 20 J/cm^2^ (Figure 2).

### 3.3. Reduction in UVA- and NaIO_3_-Induced MAPK Phosphorylation by Phlorizin

The MAPK signaling pathway is closely associated with UVA radiation, regulating various responses in RPE cells [35]. The impacts of phlorizin on the MAPK pathway activation (p-38, ERK, and JNK) triggered by UVA irradiation (20 J/cm^2^) or NaIO_3_ treatment were determined using the Western blot. Cells exposed exclusively to UVA or NAIO_3_ exhibited a heightened phosphorylation of p38, ERK, and JNK. Phlorizin, applied in escalating doses of 0.01 μM, 0.03 μM, and 0.1 μM, markedly inhibited phosphorylation levels in a dose-responsive pattern (Figure 3 and Figure 4). This highlights phlorizin’s ability to inhibit the MAPK signaling pathway and reduce inflammation induced by UVA radiation and NAIO_3_ in ARPE-19 cells.

### 3.4. Phlorizin Activation Dampens UVA- and NaIO_3_-Induced NF-κB Signal Pathway

NF-κB operates as a downstream component of the MAPK pathway and plays a critical role in the signaling cascade that governs the immune response triggered by UV radiation. It controls the expression of genes and enzymes associated with inflammation. The phosphorylation of IκB initiates the release of inflammation-related factors and influences the regulation of immune responses [66]. Western blot analysis was used to evaluate the activation of the NF-κB pathway. Both the UVA exposure and NaIO_3_ treatment significantly stimulated the NF-κB pathway activation. The pretreatment with phlorizin at escalating concentrations (0.01 μM, 0.03 μM, and 0.1 μM) significantly suppressed the phosphorylation levels of IκB and NF-κB in a dose-dependent fashion (Figure 5 and Figure 6). These data suggest that phlorizin effectively inhibits NF-κB pathways under the conditions of UVA radiation and NAIO_3_ treatment.

### 3.5. Phlorizin Downregulates Pro-Inflammatory Cytokine mRNA Expression Induced by UVA and NAIO_3_

The mRNA expression levels of the pro-inflammatory cytokines IL-1β, IL-6, and TNF-α were analyzed using RT-qPCR after exposure to the UVA radiation (20 J/cm^2^) or NAIO_3_. Both stimuli caused a significant increase in the expression of these cytokines in ARPE-19 cells. However, the pretreatment with 0.1 μM phlorizin effectively reduced the UVA-induced mRNA expression levels of IL-1β, IL-6, and TNF-α (Figure 7). The pretreatment with phlorizin at a concentration of 0.1 μM also resulted in a reduced release of IL-1β after the NAIO_3_ treatment (Figure 8). These results indicate that phlorizin mitigates the inflammatory responses induced by UVA and NAIO_3_.

### 3.6. Phlorizin Downregulates Pro-Angiogenic Factor mRNA Expression Induced by UVA and NAIO_3_

The process of angiogenesis plays a crucial role in the formation of CNV, with the VEGF identified as a major contributor to this mechanism. Beyond the VEGF, matrix metalloproteinases-2 (MMP-2) and MMP-9 have played a role in facilitating choroidal neovascularization [67]. Specifically, an increased MMP-9 mRNA expression has been detected in individuals with wet AMD [68]. Additionally, higher plasma concentrations of both MMP-2 and MMP-9 have been reported in these patients [69]. Given this background, the current experiment is designed to examine the effect of phlorizin on the expression of the pro-angiogenic factors associated with wet AMD.

The mRNA expression levels of pro-angiogenic factors VEGF, MMP2, and MMP9 were also evaluated using RT-qPCR following UVA or NaIO_3_ treatment. Both stimuli significantly upregulated the VEGF, MMP2, and MMP9 mRNA expression in ARPE-19 cells. The pretreatment with phlorizin at increasing concentrations effectively reduced the expression levels of these factors (Figure 9 and Figure 10). These findings suggest that phlorizin suppresses the pro-angiogenic factors activated by oxidative stress and inflammation.

### 3.7. Preliminary In Vivo Study: Color Fundus Photography (CFP) and Fluorescence Angiography (FA) Analysis of Effects of In Vivo Study of Phlorizin on Laser-Induced CNV (n = 3)

On Day 5, four laser photocoagulation spots were applied to each eye of the C57BL/6J mice to induce CNV formation. Phlorizin was administered intraperitoneally at varying doses (0 mg/kg, 3 mg/kg, and 10 mg/kg) starting on Day 1 as a pretreatment and continued post-laser until Day 15, marking the experiment’s conclusion. The outcomes of the CFP and FA on Day 15 are presented in Figure 11. These findings demonstrated that the pretreatment with phlorizin significantly decreased the FA leakage area on Day 15 compared to the control group.

## 4. Discussion

AMD is a multifactorial disease that significantly contributes to vision loss in the aging population. Environmental and physiological stressors, including UV radiation, blue light, oxidative stress, and inflammatory mediators, contribute to the AMD pathogenesis [10,70]. These stressors trigger excessive ROS production, leading to oxidative stress and RPE cell damage, which are central to AMD progression. Protecting RPE cells from oxidative stress is thus critical for preventing retinal diseases [71]. Phlorizin, a naturally occurring flavonoid with antioxidant and anti-inflammatory properties, has gained attention for its potential to mitigate the cellular damage induced by oxidative stress and inflammation [55,59,72,73]. This study systematically evaluated phlorizin’s protective effects on ARPE-19 cells under UVA- and NaIO_3_-induced stress, offering insights into its mechanisms and potential application for AMD.

Phlorizin exhibited no cytotoxicity in ARPE-19 cells at concentrations up to 1 μM, with MTT assay results showing a cell viability exceeding 90%. The MTT assay results demonstrated that phlorizin did not adversely affect cell viability, thereby confirming its safety for subsequent experiments. This finding aligns with the previous literature, which indicates that phlorizin exhibits minimal cytotoxicity while maintaining its bioactivity [74]. Based on these results, phlorizin concentrations of 0.01 μM, 0.03 μM, and 0.1 μM were selected for further analysis to evaluate its protective effects against UVA-induced oxidative damage.

RPE cells are vital for maintaining retinal homeostasis, as they support the recycling of photoreceptors and the transport of nutrients to the neurosensory retina [75]. The retinal environment is prone to oxidative stress due to its high levels of polyunsaturated fatty acids, significant oxygen demand, constant exposure to intense light, active mitochondrial processes, and accumulation of lipofuscin, making RPE cells especially susceptible to damage from ROS [76,77]. An overabundance of ROS can modify and impair various biomolecules, such as nucleic acids, proteins, lipids, and carbohydrates [78,79]. A previous report indicated that mitochondrial DNA damage in RPE cells was evident as a result of oxidative damage [80,81]. RPE cells in AMD donor eyes had reduced mitochondrial and glycolytic activities compared to those in non-AMD donor eyes [82]. The elevation of proliferator-activated receptor-gamma coactivator-1α (PGC-1α) and increased levels of ROS scavenging enzymes, such as catalase, copper- or zinc-containing superoxide dismutase (CuZnSOD or SOD1), and magnesium SOD, were found to be elevated in dry AMD eyes as a defense mechanism against the increased level of ROS [82,83,84]. In wet-form AMD, the total amounts of oxidants were found to be increased in the serum [85,86]. Levels of SOD, glutathione peroxidase (GPx), and glutathione reductase (GR) were also found to be lower in the serum of patients with wet-form AMD [87]. These observations indicate that oxidative stress might impair the functionality of antioxidant enzymes in individuals with the wet form of AMD. In conclusion, oxidative stress plays an important role in the pathogenesis of both wet- and dry-form AMD. UV radiation could penetrate deep into the retina and generate ROS. In this study, ARPE-19 cells exposed to increasing UVA intensities exhibited a dose-dependent reduction in viability, with a notable 38% decrease observed at 20 J/cm^2^. This observation aligns with previous research demonstrating that UVA-induced ROS overproduction causes extensive cellular damage—disrupting homeostasis, harming proteins, lipids, and DNA, impairing mitochondrial function, and ultimately leading to cell death through apoptosis or necrosis [9,33,88,89,90,91]. The phlorizin pretreatment enhanced ARPE-19 cells’ viability in a concentration-dependent manner. Specifically, at a concentration of 0.1 μM, phlorizin restored cell viability to approximately 85%. This protective effect arises from phlorizin’s ability to significantly reduce UVA-induced ROS production, as shown in Figure 2. These findings are consistent with prior studies, which indicate that phlorizin effectively lowers ROS formation, thereby safeguarding RPE cells from oxidative damage [92,93].

The MAPK signaling pathway, comprising p38, ERK, and JNK proteins, plays a pivotal role in mediating cellular responses to oxidative stress and inflammation [94]. The activation of these kinases under UVA exposure leads to downstream signaling events that exacerbate inflammation and apoptosis. The Western blot analysis in this study revealed that the UVA- and NaIO_3_- induced phosphorylation of p38, ERK, and JNK was significantly attenuated by the phlorizin pretreatment. The inhibitory effect was dose-dependent, suggesting that phlorizin effectively modulates the MAPK pathway activation to reduce oxidative stress and inflammation. In addition to MAPK signaling, the NF-κB pathway is another critical mediator of inflammation and oxidative stress in RPE cells [95,96]. The activation of NF-κB leads to the transcription of pro-inflammatory cytokines, such as IL-1β, IL-6, and TNF-α, as well as pro-angiogenic factors, like the VEGF and matrix metalloproteinases (MMPs) [8,97]. TNF-α was identified as a key factor in macrophage-driven ocular angiogenesis, with macrophages playing an essential role in the effective remodeling of retinal vasculature and the process of neovascularization [22,98,99,100]. Increasing macrophages in a damaged Bruch’s membrane promotes the release of TNF-α and ILs [101]. IL-6 is associated with the induction of ocular inflammation and angiogenesis [102]. Higher blood concentrations of IL-6 and IL-1β were found in patients with wet AMD and were associated with disease severity [102,103]. The VEGF also plays an important role in pathological ocular angiogenesis [104,105]. Our study demonstrated that the UVA-induced phosphorylation of IκB and NF-κB was significantly reduced in phlorizin-pre-treated ARPE-19 cells. The RT-qPCR analysis further confirmed that the phlorizin pretreatment significantly reduced the UVA-induced mRNA expression of pro-inflammatory cytokines (IL-1β, IL-6, and TNF-α) and pro-angiogenic factors (VEGF, MMP-2, and MMP-9). The inhibitory effect was dose-dependent, emphasizing phlorizin’s ability to modulate signaling pathways and suppress inflammatory responses at the transcriptional level. In addition, the in vivo study revealed that the phlorizin treatment significantly reduced the leakage area and FA score, indicating that phlorizin may be effective in reducing leakage from neovascularization and may be a potential treatment option for neovascular AMD.

There were two clinical studies, Age-Related Eye Disease Study (AREDS) and AREDS2, that revealed that the dietary intake of antioxidants mitigated the progression of AMD [106,107]. These two studies revealed that dietary supplements of vitamin C, vitamin E, zinc, lutein, zeaxanthin, and beta carotene may be beneficial for patients with AMD. Other natural compounds, including resveratrol and cerium oxide, have been reported to effectively protect RPE cells from oxidative stress. This highlights the potential of natural compounds as therapeutic agents in combating oxidative damage [108,109,110,111,112,113]. Flavonoids, a class of plant-derived compounds, have been widely studied for their protective effects against various eye disorders, including AMD, cataracts, glaucoma, and diabetic retinopathy [114,115]. Notably, a study by Gopinath Bamini et al. demonstrated that a higher flavonoid consumption was related to a reduced risk of AMD development [116]. These benefits stem from the ability of flavonoids to reduce oxidative stress, inflammation, and angiogenesis [117]. Other notable flavonoids that may be helpful in preventing or treating AMD are quercetin and luteolin. Quercetin are rich in red wine, apples, onions, and berries [118]. Quercetin shields RPE cells from oxidative stress and helps prevent retinal degeneration [119]. It achieves this by modulating signaling pathways, like MAPK/ERK and cAMP Response Element-Binding Protein (CREB), which lower the levels of pro-inflammatory cytokines, such as IL-6, IL-8, and Monocyte Chemoattractant Protein-1 (MCP-1) [120]. Additionally, quercetin protects ARPE cells from the oxidative stress induced by NaIO_3_ through modulating the acetylation of SOD2 via the Nrf2-PGC-1α-Sirt1 signaling pathway [121]. Similarly, luteolin could protect RPE cells from oxidative stress and significantly reduce the levels of IL-6, IL-8, and MCP-1 [120]. Luteolin also inhibited the activation of the MAPK and NF-κB pathways in IL-1β stimulated ARPE cells [122]. Despite these benefits, luteolin’s therapeutic use for AMD is limited by its poor aqueous and lipid solubility and low bioavailability [117]. Other flavonoids, such as wogonin and kaempferol, have also shown similar protective effects in ARPE cells [123,124]. Phlorizin is also considered an antioxidant and previous studies have revealed its potential for alleviating conditions associated with oxidative stress. For instance, phlorizin has been shown to protect neural cells by alleviating oxidative stress through the activation of the Nrf2 pathway and reducing NF-κB-mediated inflammatory responses [55]. Additionally, its ability to inhibit UVB-induced apoptosis in HaCaT keratinocytes further supports its potential as a photoprotective agent [125]. In our study, we found that the phlorizin treatment significantly reduced the activation of the NF-κB and MAPK pathways in ARPE cells, leading to a decrease in the release of pro-inflammatory cytokines and pro-angiogenic factors. Our study highlighted the therapeutic potential of phlorizin in AMD.

However, this study has limitations that affect its translational application to human AMD. The ARPE-19 cell line, while widely used, may not fully mimic the responses of primary RPE cells under oxidative stress or in response to phlorizin treatments. Similarly, the laser-induced choroidal neovascularization (CNV) mouse model, though effective for studying neovascularization, is primarily suited for investigating wet AMD, limiting its applicability to dry AMD. Furthermore, confirming the effective intraocular concentrations of phlorizin (0.01–0.1 μM) through pharmacokinetic studies and clinical trials is essential to bridge the gap between these preclinical findings and clinical applications.

To the best of our knowledge, this study represents the first comprehensive investigation into the potential mechanisms underlying phlorizin and UVA-induced damage to ARPE cells. Additionally, it explores the potential therapeutic applications of phlorizin in the treatment of AMD. Future studies are needed to validate the potential of phlorizin as a novel treatment option for AMD.

## 5. Conclusions

In conclusion, this study demonstrates that phlorizin effectively protects ARPE-19 cells from UVA- and NaIO3-induced oxidative stress and inflammation by inhibiting ROS production and modulating the MAPK and NF-κB signaling pathways (Figure 12). The phlorizin treatment reduced the expression of pro-inflammatory cytokines and pro-angiogenic factors. The in vivo study using a laser CNV model in C57BL/6J mice revealed that the phlorizin treatment significantly reduced the CNV leakage and FA leakage score. Phlorizin, a natural compound known for its high bioavailability and low toxicity, shows promise as a safe and effective intervention for preventing or mitigating the progression of AMD. Yet, translating these findings into clinical applications may face challenges, including determining optimal human dosing, assessing phlorizin’s bioavailability in ocular tissues, and evaluating the long-term safety in humans. Future research should address these limitations to facilitate the development of phlorizin-based therapies for age-related retinal diseases.

## Figures and Tables

**Figure 1 biomolecules-15-00523-f001:**
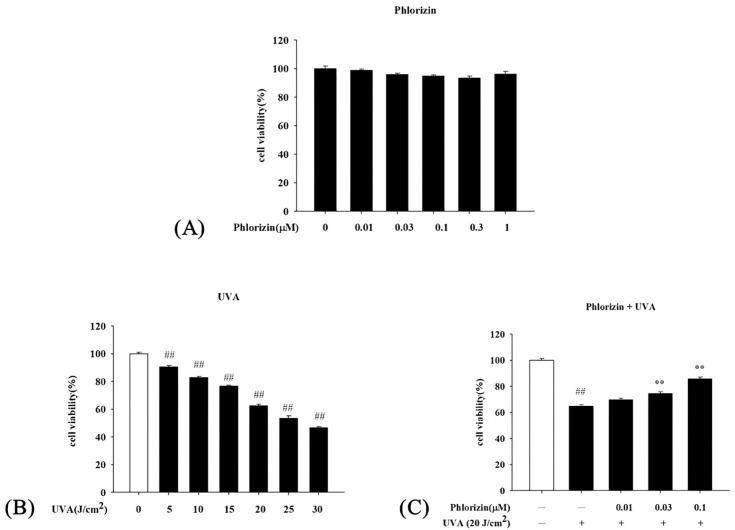
Cell Viability Test for ARPE-19 cells. (**A**) ARPE-19 cells were treated with phlorizin at varying concentrations (0, 0.01, 0.03, 0.1, 0.3, and 1 μM) for 24 h in 5% CO_2_ atmosphere, and cell viability was determined with MTT assay. (**B**) UVA radiation was applied at different doses (5, 10, 15, 20, 25, and 30 J/cm^2^) followed by 24 h incubation period, with cell viability measured with MTT assay. (**C**) Cells were pre-incubated with varying phlorizin concentrations for 24 h, then exposed to 20 J/cm^2^ UVA radiation and incubated for additional 24 h; cell viability was evaluated using MTT assay. Data are expressed as mean ± SEM from minimum of three independent experiments. ## *p* < 0.01 compared to control group; ** *p* < 0.01 compared to phlorizin-negative/UVA-positive group.

**Figure 2 biomolecules-15-00523-f002:**
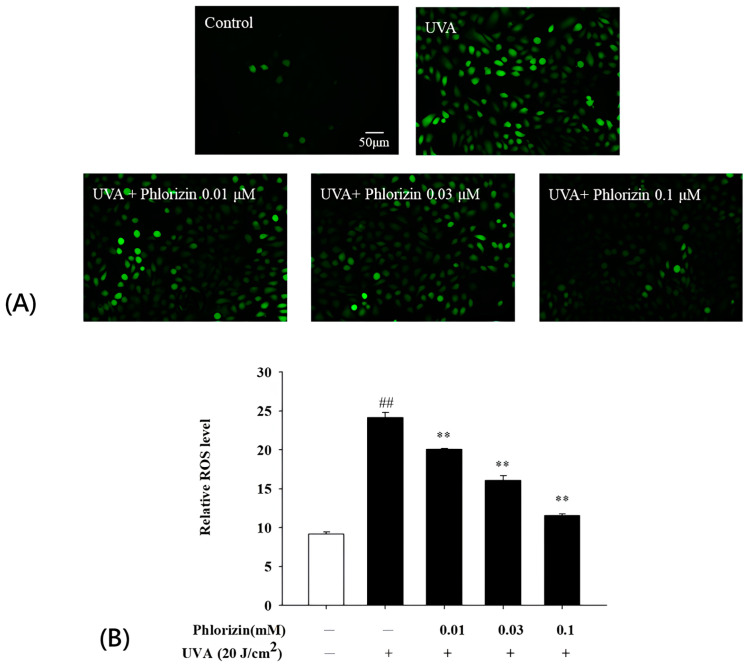
Phlorizin inhibits ROS production induced by UVA radiation. (**A**) Cells were pre-treated with different doses of phlorizin for 24 h, followed by addition of DCFH-DA dye and 30 min incubation period. Subsequently, cells were irradiated with UVA at 20 J/cm^2^. Post-irradiation, medium was replaced with fresh medium, and cells were incubated for 15 min prior to fluorescence measurement. (**B**) ROS levels were assessed in ARPE-19 cells pre-treated with different phlorizin concentrations for 24 h and then exposed to UVA radiation at 20 J/cm^2^. Data are presented as mean ± SEM from minimum of five independent experiments. ## *p* < 0.01 compared to control group; ** *p* < 0.01 compared to phlorizin-negative/UVA-positive group.

**Figure 3 biomolecules-15-00523-f003:**
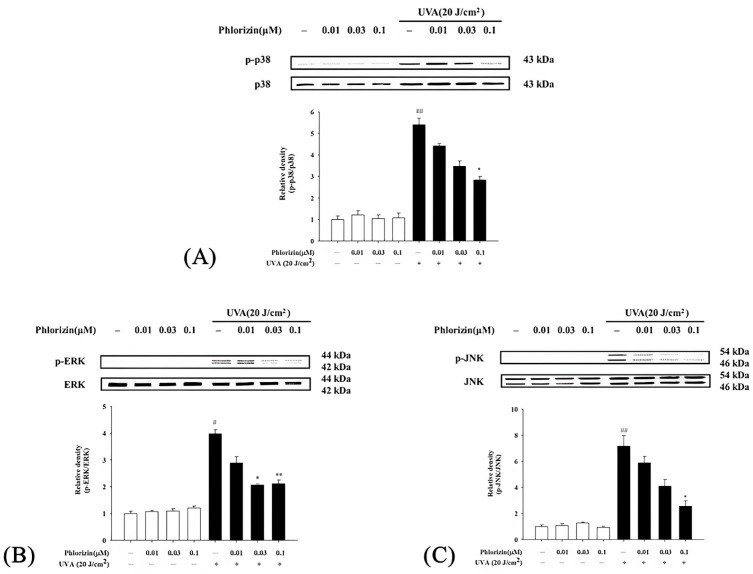
Effect of phlorizin treatment on UVA-induced MAPK pathway activation in ARPE-19 cells. (**A**) p38, (**B**) ERK, and (**C**) JNK were analyzed. Cells were pre-treated with different doses of phlorizin for 24 h, and then cells were exposed to UVA (20 J/cm^2^) radiation. After UVA exposure, fresh medium was added and after additional 30 min of incubation, total protein was extracted from cells, and P38, ERK, and JNK expression were determined using Western blotting. Data are expressed as mean ± SEM from minimum of five independent experiments. ## *p* < 0.01 and # *p* < 0.05 compared to control group; ** *p* < 0.01 and * *p* < 0.05 compared to group treated with phlorizin-negative/UVA positive group.

**Figure 4 biomolecules-15-00523-f004:**
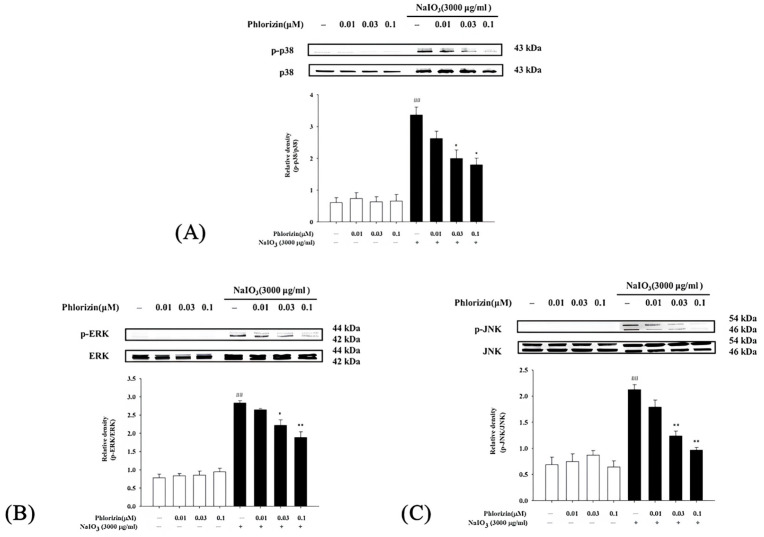
Effect of phlorizin treatment on NaIO₃-induced MAPK pathway activation in ARPE-19 cells. (**A**) p38, (**B**) ERK, and (**C**) JNK levels were evaluated. Cells were pre-treated with different doses of phlorizin for 24 h, followed by exposure to NaIO₃ (3000 μg/mL) for additional 24 h. Following NaIO₃ treatment, medium was replaced, and after additional 30 min incubation period, total protein was extracted from cells. Expression levels of p38, ERK, and JNK were assessed via Western blot analysis. Data are reported as mean ± SEM from minimum of five independent experiments. ## *p* < 0.01 compared to control group; ** *p* < 0.01 and * *p* < 0.05 compared to group treated with phlorizin negative/NaIO_3_ positive group.

**Figure 5 biomolecules-15-00523-f005:**
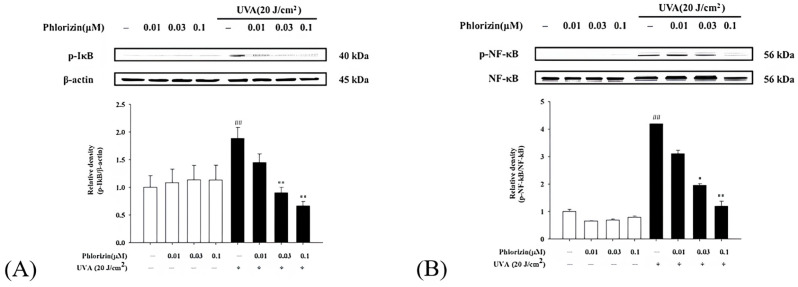
Effect of phlorizin treatment on UVA-induced NF-κB pathway activation in ARPE-19 cells. (**A**) IκB and (**B**) NF-κB levels were examined. Cells were pre-treated with different doses of phlorizin for 24 h, followed by exposure to UVA radiation (20 J/cm^2^). Following UVA exposure, medium was substituted with fresh medium, and cells were further incubated for one hour. Total protein was then extracted, and expression of IκB and NF-κB was assessed using Western blot analysis. Data are presented as mean ± SEM from at least five independent experiments. ## *p* < 0.01 vs. control group; ** *p* < 0.01 and * *p* < 0.05 vs. the phlorizin-negative/UVA-positive group.

**Figure 6 biomolecules-15-00523-f006:**
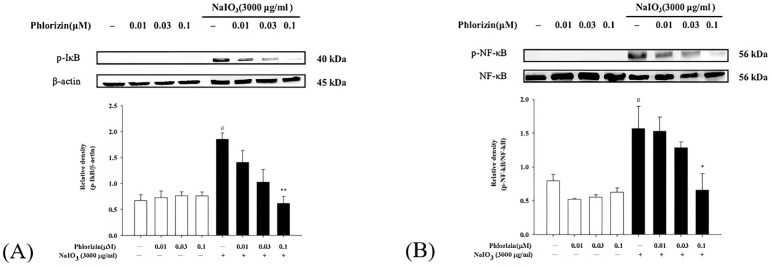
Effect of phlorizin treatment on NaIO_3_-induced NF-κB pathway activation in ARPE-19 cells. (**A**) IκB and (**B**) NF-κB levels were investigated. Cells were pre-treated with different doses of phlorizin for 24 h, followed by treatment with NaIO_3_ for additional 24 h. Following NaIO_3_ treatment, medium was replaced, and after additional 30 min incubation period, total protein was extracted from the cells. Total protein was subsequently extracted, and expression of IκB and NF-κB was evaluated using Western blot analysis. Data are expressed as mean ± SEM from minimum of five independent experiments. # *p* < 0.05 vs. control group; ** *p* < 0.01 and * *p* < 0.05 vs. the phlorizin-negative/NAIO_3_-positive group.

**Figure 7 biomolecules-15-00523-f007:**
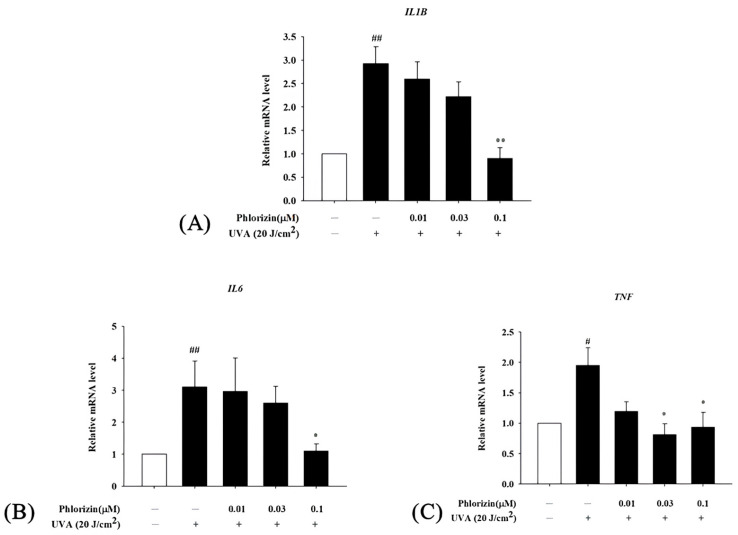
Effect of phlorizin on levels of inflammatory cytokines after UVA radiation in ARPE-19 cells. (**A**) IL-1β, (**B**) IL6, and (**C**) TNF-α. Cells were pre-treated with different doses of phlorizin for 24 h prior to being irradiated with UVA radiation at dose of 20 J/cm^2^. After UVA exposure, fresh medium was added and after additional 2 h of incubation total RNA was extracted, and mRNA expression levels were quantified using real-time RT-PCR. Data are presented as mean ± SEM from minimum of five independent experiments. ## *p* < 0.01 and # *p* < 0.05 vs. control group; ** *p* < 0.01; and * *p* < 0.05 vs. phlorizin- negative/UVA-positive group.

**Figure 8 biomolecules-15-00523-f008:**
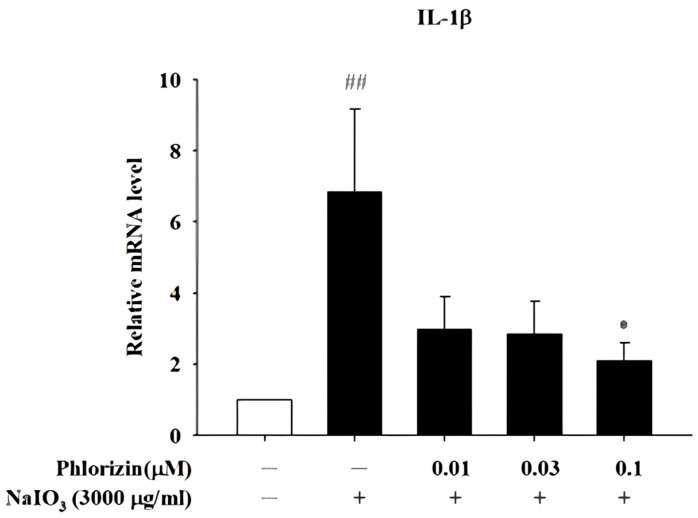
Effect of phlorizin on levels of IL-1β after NAIO_3_ treatment in ARPE-19 cells. Cells were pre-treated with different doses of phlorizin for 24 h and then stimulated with NAIO_3_ for 3 h. Total RNA was isolated, and mRNA expression levels were measured via real time RT-PCR. Data are presented as mean ± SEM from minimum of five independent experiments ## *p* < 0.01 vs. control group; * *p* < 0.05, compared with phlorizin-negative/NAIO_3_—positive group.

**Figure 9 biomolecules-15-00523-f009:**
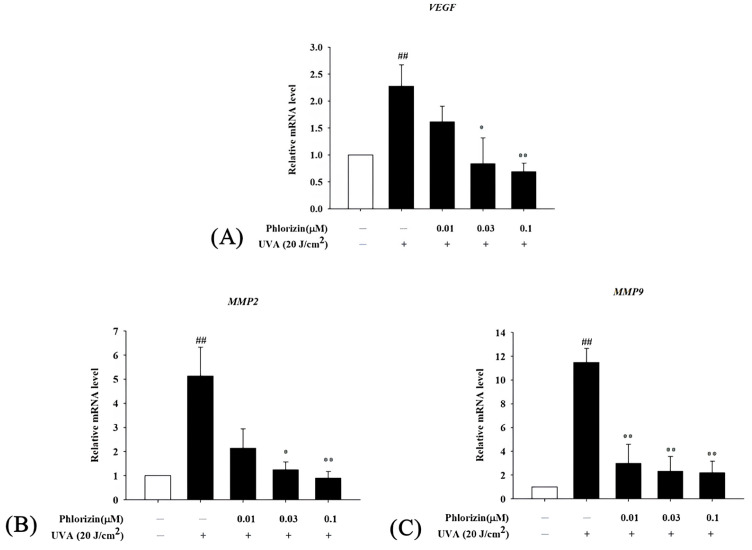
Effect of phlorizin on levels of proangiogenic factors after UVA radiation in ARPE-19 cells. (**A**) VEGF, (**B**) MMP2, and (**C**) MMP9. Cells were pre-treated with different doses of phlorizin for 24 h, and then cells were exposed to UVA (20 J/cm^2^) radiation. After UVA exposure, fresh medium was added and after additional 2 h of incubation total RNA was extracted, and mRNA expression levels were quantified using real-time RT-PCR. Data are presented as mean ± SEM from minimum of five independent experiments ## *p* < 0.01 vs. control group; * *p* < 0.05, ** *p* < 0.01 vs. phlorizin-negative/UVA-positive group.

**Figure 10 biomolecules-15-00523-f010:**
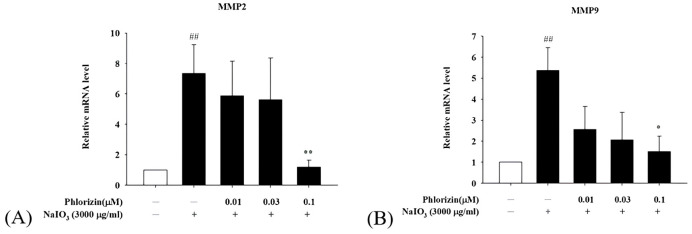
Effect of phlorizin on levels of proangiogenic factors after NAIO_3_ treatment in ARPE-19 cells. (**A**) MMP2 and (**B**) MMP9. Cells were pre-treated with different doses of phlorizin for 24 h and then stimulated with NAIO_3_ (3000 μg/mL) for 24 h. Total RNA was extracted, and mRNA expression levels were quantified using real-time RT-PCR. Data are presented as mean ± SEM from minimum of five independent experiments. ## *p* < 0.01 vs. control group; ** *p* < 0.01 and * *p* < 0.05 vs. phlorizin-negative/NAIO_3_–positive group.

**Figure 11 biomolecules-15-00523-f011:**
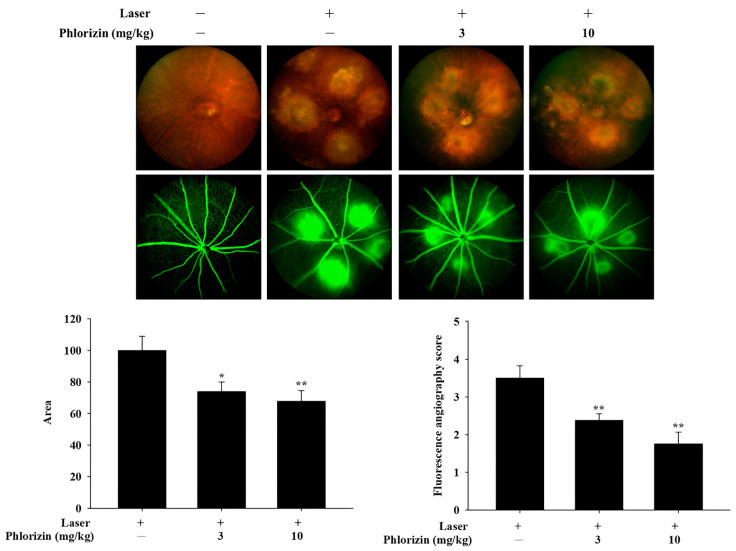
CFP and FA analysis on Day 15 (10 days after laser photocoagulation). Results revealed significantly lower leakage area and lower FA score in phlorizin pretreatment group (3 mg/kg and 10 mg/kg) compared to Laser +/Phlorizin—group. * *p* < 0.05; ** *p* < 0.01, compared to Laser +/Phlorizin—group.

**Figure 12 biomolecules-15-00523-f012:**
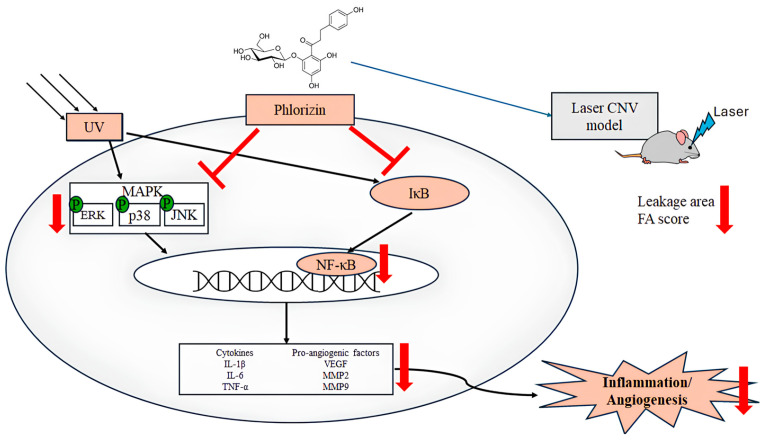
Mechanism of phlorizin under ultraviolet radiation-induced inflammation and angiogenesis.

**Table 1 biomolecules-15-00523-t001:** Human primer sequence for RT-PCR.

Genes	Primers	Sequence (5′-3′)
GAPDH	Forward	GGGGCTGCCCAGAACATCAT
	Reverse	GCCTGCTTCACCACCTTCTTG
IL1-β	Forward	GCTGAGGAAGATGCTGGTTC
	Reverse	TCCATATCCTGTCCCTGGAG
IL-6	Forward	CACAGACAGCCACTCACCTC
	Reverse	TTTTCTGCCAGTGCCTCTTT
IL-8	Forward	GCAGAGGGTTGTGGAGAAGT
	Reverse	TGGCATCTTCACTGATTCTTGG
TNF-α	Forward	ATGCAGTTTGGCCAAGGAGA
	Reverse	GCTTCTCAACAACCCTCTGA
VEGF	Forward	AGTTCCACCACCAAACATGC
	Reverse	TGAAGGGACACAACGACACA
MMP-2	Forward	TGGCAAGGTGTGGTGTGCGAC
	Reverse	TCGGGGCCATCAGAGCTCCAG
MMP-9	Forward	GGTGTGCCCTGGAACTCACACG
	Reverse	AGGGCACTGCAGGAGGTCGT

## Data Availability

The original contributions presented in this study are included in the article/Appendix A. Further inquiries can be directed to the corresponding author(s).

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
