# Peer review of "Phlorizin Protects Against Oxidative Stress and Inflammation in Age-Related Macular Degeneration Model"

_biomolecules, 2025, doi:10.3390/biom15040523_

Round 1

Reviewer 1 Report

Comments and Suggestions for Authors

The authors of the article ID: 3513531, title: Phlorizin, the Main Component of Sweet Tea, Protects Against Oxidative Stress and Inflammation in Age-Related Macular Degeneration Models have submitted interesting research results for publication in the journal Biomolecules.

In this study investigated the therapeutic potential sweet tea (Lithocarpus polystachyus Rehd.) of phlorizin in age-related macular degeneration (AMD) models. The authors showed that phlorizin demonstrated significant protective effects against oxidative stress and inflammation, highlighting its therapeutic potential for treating AMD.

The authors used many research tools, such as ARPE-19 retinal pigment epithelial cell line, MTT test, western blotting, measurement of ROS, Real-time Quantitative Polymerase Chain Reaction (RT-qPCR) Analysis, Fluorescence Angiography etc., which indicates good skills and provides the possibility of good research.

Statistical comparisons between control and experimental groups were performed using unpaired two-tailed. Student’s t-tests via SigmaPlot software.

The research results are well prepared both in tables and figures. The research results are widely commented on in well-matched literature. A total of 119 literature items were presented in the introduction and discussion.

Comments on the Quality of English Language

In my opinion, the manuscript is prepared at a high level.

Author Response

Comments and Suggestions for Authors

The authors of the article ID: 3513531, title: Phlorizin, the Main Component of Sweet Tea, Protects Against Oxidative Stress and Inflammation in Age-Related Macular Degeneration Models have submitted interesting research results for publication in the journal Biomolecules.

In this study investigated the therapeutic potential sweet tea (Lithocarpus polystachyus Rehd.) of phlorizin in age-related macular degeneration (AMD) models. The authors showed that phlorizin demonstrated significant protective effects against oxidative stress and inflammation, highlighting its therapeutic potential for treating AMD.

The authors used many research tools, such as ARPE-19 retinal pigment epithelial cell line, MTT test, western blotting, measurement of ROS, Real-time Quantitative Polymerase Chain Reaction (RT-qPCR) Analysis, Fluorescence Angiography etc., which indicates good skills and provides the possibility of good research.

Statistical comparisons between control and experimental groups were performed using unpaired two-tailed. Student’s t-tests via SigmaPlot software. 

The research results are well prepared both in tables and figures. The research results are widely commented on in well-matched literature. A total of 119 literature items were presented in the introduction and discussion.

 Comments on the Quality of English Language

In my opinion, the manuscript is prepared at a high level.

Reply:

Thank you for taking the time to review our manuscript and for your encouraging and positive feedback. We believe that your feedback has been fully addressed in the current version of the manuscript, and we hope that it meets the standards for publication in Biomolecules. Should you have any additional suggestions, we would be happy to consider them. Thank you once again for your time, thoughtful review, and valuable input. Your comments have been instrumental in affirming the quality of our work.

Reviewer 2 Report

Comments and Suggestions for Authors

The paper by Zhen-Yu Liao et al is interesting and well formulated after minor revision

Author Response

The authors write in Abstract

Given the critical role of oxidative stress and inflammation in age-related macular degeneration (AMD), this study investigated the therapeutic potential of phlorizin in AMD models.

Consider specifying the hypothesis or research question to provide a clearer focus for the study.

Reply: We agree that explicitly stating the hypothesis enhances the Abstract’s focus. We have revised the Abstract to include a clear hypothesis, linking it directly to the study’s objectives and experimental design.

The authors write in Introduction

Despite advances in treatments, including anti-VEGF therapy, novel preventive and therapeutic strategies are needed for both dry and wet AMD.

You might add a brief mention of why current therapies are insufficient to further emphasize the need for new treatments.

Reply: We appreciate this suggestion and have modified the Introduction to briefly explain the limitations of current therapies, thereby strengthening the rationale for investigating phlorizin as a novel approach.

Moreover ..

Phlorizin, a flavonoid derived from sweet tea and apple tree bark, possesses antioxidative, antiinflammatory, anti-tumor, and neuroprotective properties.

It may be beneficial to provide a reference or brief explanation of previous studies supporting these claims.

Reply: We appreciate this suggestion, and we have briefly introduced the effects of phlorizin in revised manuscripts.

The authors write in conclusion

Phlorizin effectively protects ARPE-19 cells from oxidative damage and inflammation, offering a promising approach for AMD treatment. Future research should explore its clinical translation and potential as a dietary supplement for AMD prevention.

You could discuss potential limitations or challenges in translating these findings into clinical applications.

Reply: Thank you for your feedback and we have added limitations in the Discussion section.

Reviewer 3 Report

Comments and Suggestions for Authors

Review of the article “Phlorizin, the Main Component of Sweet Tea, Protects Against Oxidative Stress and Inflammation in Age-Related Macular Degeneration Models” ,submitted to Biomolecules. The study presents an original and well-executed investigation of the potential protective effects of phlorizin against oxidative stress and inflammation in in vitro and in vivo models of age-related macular degeneration (AMD).

The text is written correctly in English – the scientific style, logical coherence, and technical precision are at a good level. However, there are minor grammatical and syntactic errors (e.g. repetitions, too long sentences, or too frequent use of passives). There are also occasional lack of conjunctions or logical transitions between paragraphs. These corrections do not significantly affect the understanding of the content, but they are worth including in the final version.

Below are detailed comments on the peer-reviewed article:

Strengths of the manuscript:

  1. AMD remains a pressing public health concern, and the exploration of naturally derived compounds, such as phlorizin, introduces a potentially novel therapeutic strategy. The title suggests that the article concerns the protective effect of phlorizin against oxidative stress and inflammation in AMD models. The authors have indeed conducted a comprehensive analysis:
  • Cellular models (ARPE-19, UVA irradiation, NaIO₃),
  • ROS, MAPK, NF-κB analyses,
  • measurements of cytokines and angiogenic factors,
  • In vivo studies in a mouse model with induced CNV.

Therefore, the issues in the title have been discussed in great detail and exhaustively.

  1. The methodology is solid, employing both in vitro (ARPE-19 cells) and in vivo (CNV mouse model) approaches to validate the findings.

The manuscript is well organised and clearly written. The objectives, methods, results, and conclusions are coherent and well presented. The topic is definitely current and important – AMD is one of the most common neurodegenerative eye diseases in the elderly, and the growing number of cases increases the need for alternative therapies.

The novelty of the article is the following.

  • The use of phlorizin, so far poorly studied in the context of AMD.
  • Comprehensive analysis of molecular pathways (MAPK, NF-κB).
  • Confirmation of results in an animal model (laser-induced CNV).

The authors claim that this is the first such comprehensive study of the effect of phlorizin on AMD models, which, looking at the literature, seems accurate.

  1. The literature cited is mostly current (within the last 5 years), demonstrating awareness of recent developments in the field. The references are accurate, reliable, and represent the current state of knowledge. There are also classic citations from earlier years (eg, 2003-2015), which is historically justified.

Minor recommendations:

  1. Some sentences in the discussion are somewhat redundant and could benefit from slight condensation for clarity and focus.
  2. Although the role of phlorizin is well known here, it would be beneficial to briefly compare its activity with other flavonoids previously studied in AMD (eg quercetin, lutein) to better contextualise its therapeutic potential.
  3. The authors are encouraged to elaborate more clearly on the limitations of their study, especially in terms of translational relevance to human models.

Conclusions

In my opinion, this manuscript provides a significant contribution to the field of research on retinal disease and natural compound-based therapies. The work is comprehensive, methodologically robust, and well supported by experimental data. After addressing the minor issues listed above, the manuscript will be suitable for publication.

Recommendations: Minor revision

Comments on the Quality of English Language

The text is written correctly in English – the scientific style, logical coherence, and technical precision are at a good level. However, there are minor grammatical and syntactic errors (e.g. repetitions, too long sentences, or too frequent use of passives). There are also occasional lack of conjunctions or logical transitions between paragraphs. These corrections do not significantly affect the understanding of the content, but they are worth including in the final version.

Author Response

Comments and Suggestions for Authors

Review of the article “Phlorizin, the Main Component of Sweet Tea, Protects Against Oxidative Stress and Inflammation in Age-Related Macular Degeneration Models”, submitted to Biomolecules. The study presents an original and well-executed investigation of the potential protective effects of phlorizin against oxidative stress and inflammation in in vitro and in vivo models of age-related macular degeneration (AMD).

The text is written correctly in English – the scientific style, logical coherence, and technical precision are at a good level. However, there are minor grammatical and syntactic errors (e.g. repetitions, too long sentences, or too frequent use of passives). There are also occasional lack of conjunctions or logical transitions between paragraphs. These corrections do not significantly affect the understanding of the content, but they are worth including in the final version.

Below are detailed comments on the peer-reviewed article:

Strengths of the manuscript:

  1. AMD remains a pressing public health concern, and the exploration of naturally derived compounds, such as phlorizin, introduces a potentially novel therapeutic strategy. The title suggests that the article concerns the protective effect of phlorizin against oxidative stress and inflammation in AMD models. The authors have indeed conducted a comprehensive analysis:
  • Cellular models (ARPE-19, UVA irradiation, NaIO₃),
  • ROS, MAPK, NF-κB analyses,
  • measurements of cytokines and angiogenic factors,
  • In vivo studies in a mouse model with induced CNV.

Therefore, the issues in the title have been discussed in great detail and exhaustively.

  1. The methodology is solid, employing both in vitro (ARPE-19 cells) and in vivo (CNV mouse model) approaches to validate the findings.

The manuscript is well organised and clearly written. The objectives, methods, results, and conclusions are coherent and well presented. The topic is definitely current and important – AMD is one of the most common neurodegenerative eye diseases in the elderly, and the growing number of cases increases the need for alternative therapies.

The novelty of the article is the following.

  • The use of phlorizin, so far poorly studied in the context of AMD.
  • Comprehensive analysis of molecular pathways (MAPK, NF-κB).
  • Confirmation of results in an animal model (laser-induced CNV).

The authors claim that this is the first such comprehensive study of the effect of phlorizin on AMD models, which, looking at the literature, seems accurate.

  1. The literature cited is mostly current (within the last 5 years), demonstrating awareness of recent developments in the field. The references are accurate, reliable, and represent the current state of knowledge. There are also classic citations from earlier years (eg, 2003-2015), which is historically justified.

Minor recommendations:

  1. Some sentences in the discussion are somewhat redundant and could benefit from slight condensation for clarity and focus.

Reply: Thank you for your opinion and we have made relevant revisions to the discussion section to improve clarity and prevent redundancy.

  1. Although the role of phlorizin is well known here, it would be beneficial to briefly compare its activity with other flavonoids previously studied in AMD (eg quercetin, lutein) to better contextualise its therapeutic potential.

Reply: We have added discussion on the effects of other flavonoids and possible mechanisms in the discussion section. Thank you.

  1. The authors are encouraged to elaborate more clearly on the limitations of their study, especially in terms of translational relevance to human models.

Reply: Thank you for your insightful opinion. We have added limitations in the Discussion section.

Conclusions

In my opinion, this manuscript provides a significant contribution to the field of research on retinal disease and natural compound-based therapies. The work is comprehensive, methodologically robust, and well supported by experimental data. After addressing the minor issues listed above, the manuscript will be suitable for publication.

Reply: Thank you for your positive feedback and we have made changes as requested.

Recommendations: Minor revision

Comments on the Quality of English Language

The text is written correctly in English – the scientific style, logical coherence, and technical precision are at a good level. However, there are minor grammatical and syntactic errors (e.g. repetitions, too long sentences, or too frequent use of passives). There are also occasional lack of conjunctions or logical transitions between paragraphs. These corrections do not significantly affect the understanding of the content, but they are worth including in the final version.

Reply: We have made relevant changes to the manuscript to improve clarity and reduce redundancy.